# Is Dairy Effluent an Alternative for Maize Crop Fertigation in Semiarid Regions? An Approach to Agronomic and Environmental Effects

**DOI:** 10.3390/ani12162025

**Published:** 2022-08-10

**Authors:** Banira Lombardi, Luciano Orden, Patricio Varela, Maximiliano Garay, Gastón Alejandro Iocoli, Agustín Montenegro, José Sáez-Tovar, María Ángeles Bustamante, María Paula Juliarena, Raul Moral

**Affiliations:** 1CIFICEN (CONICET-UNCPBA-CICPBA), Pinto 399, Tandil 7000, Argentina; 2Estación Experimental Agropecuaria INTA Ascasubi, Ruta 3 Km 794, 8142, Hilario Ascasubi, Buenos Aires 8000, Argentina; 3Departamento de Agronomía, Universidad Nacional del Sur (UNS), San Andrés 800, Bahía Blanca 8000, Argentina; 4Centro de Investigación e Innovación Agroalimentaria y Agroambiental (CIAGRO-UMH), Universidad Miguel Hernández, Carretera de Beniel Km 3, 2, 03312 Orihuela, Spain; 5Facultad de Ciencias Exactas-UNCPBA, Pinto 399, Tandil 7000, Argentina

**Keywords:** *Zea mays* L., livestock effluent, methane, nitrous oxide, carbon dioxide, subsurface drip fertigation

## Abstract

**Simple Summary:**

Dairy effluent can be an environmental problem if it is not properly managed. Several application technologies exist for its reuse as a source of nutrients in agricultural crops. Our study provides new information on GHG emissions after the application of dairy effluent through a subsurface drip irrigation system on a semiarid soil of the southern Pampean Region (Argentina). In addition, some edaphic properties are compared with conventional chemical fertilization on the yield of a corn crop, contributing as a proposal for an improvement in agricultural sustainability.

**Abstract:**

The reuse of effluents from intensive dairy farms combined with localized irrigation techniques (fertigation) has become a promising alternative to increase crop productivity while reducing the environmental impact of waste accumulation and industrial fertilizers production. Currently, the reuse of dairy effluents through fertigation by subsurface drip irrigation (SDI) systems is of vital importance for arid regions but it has been poorly studied. The present study aimed to assess the greenhouse gas (GHG) emissions, soil properties, and crop yield of a maize crop fertigated with either treated dairy effluent or dissolved granulated urea applied through an SDI system at a normalized N application rate of 200 kg N ha^−1^. Fertilizer application was divided into six fertigation events. GHG fluxes were measured during fertigation (62-day) using static chambers. Soil properties were measured previous to fertilizer applications and at the harvest coinciding with crop yield estimation. A slight increase in soil organic matter was observed in both treatments for the 20–60 cm soil depth. Both treatments also showed similar maize yields, but the dairy effluent increased net GHG emissions more than urea during the fertigation period. Nevertheless, the net GHG emissions from the dairy effluent were lower than the theoretical CO_2_eq emission that would have been emitted during urea manufacturing or the longer storage of the effluent if it had not been used, showing the need for life-cycle assessments. Local-specific emission factors for N_2_O were determined (0.07%), which were substantially lower than the default value (0.5%) of IPCC 2019. Thus, the subsurface drip irrigation systems can lead to low GHG emissions, although further studies are needed.

## 1. Introduction

Due to the burgeoning world population, but mainly due to improvements in the population’s standard of living, the demand for agricultural products is expected to increase by around 70% by 2050 [1,2]. In this scenario, the intensification of animal production and the increase in the size of production units are the main trends in livestock activity [3,4]. This will generate a significant increase in effluents and organic wastes that could result in potential contaminants if not managed properly [5]. There are specific problems with cattle waste management, such as odors, pathogens, water and soil contamination, and ammonia and greenhouse gas (GHG) emissions, such as nitrous oxide (N_2_O), methane (CH_4_), and carbon dioxide (CO_2_) [6]. On dairy farms, for example, there are several areas which are potential sources of GHG. In fact, and according to a review by Owen and Silver [7], anaerobic lagoons are sources of CH_4_ and N_2_O that exceed direct CH_4_ emissions from dairy cows.

Dairy effluents contain significant amounts of organic matter and nutrients such as carbon (C) and nitrogen (N) in reduced forms and, therefore, they are rich in energy. After treatment, the liquid manure can be used as a soil amendment in order to improve plant growth and soil quality. At the same time, the use of these amendments would increase the sustainability of farming systems [8], since the negative impacts of their continuous accumulation in livestock production would be reduced [9]. In addition, the use of limited resources in the production of synthetic fertilizers would be minimized [3,10].

On the other hand, the incorporation of fertilizers (either organic or inorganic) with high ammonia (NH_4_^+^) concentration promotes nitrification and O_2_ consumption (CO_2_ release), generating suboxic conditions. Thus, nitrite (NO_2_^−^) accumulation promotes that it is reduced to N_2_O, a process called nitrifier denitrification [11,12]. In addition, the application of labile organic compounds increases respiration, promoting anaerobic conditions and the degradation of organic matter with CH_4_ release [13]. In addition, in these conditions, NO_3_^−^ is used as an electron acceptor with NO_2_^−^, N_2_O, or N_2_ release depending on the metabolic pathway. Irrigation also promotes suboxic conditions, favoring the previously described emissions.

Irrigation with treated wastewater is already being implemented in many countries, mainly for agriculture and landscaping [14]. This strategy is especially important for soils of arid regions where water is a limited resource due to scarce rainfall and typically high evaporation [15]. These arid zones are of great importance to our planet not only because they occupy 40% of the Earth’s total surface but also because they provide much of the world’s cereals and livestock [16]. Moreover, the change in land use for agricultural purposes in dry areas modifies the quantity and quality of organic matter, reducing the reserve of C and other soil nutrients, especially N [17]. In this context, using nonconventional water resources, for example, the reuse of wastewater, appears as an alternative to satisfy, at least partially, the demand for water and nutrients, giving added value to waste products [18]. Although one of the main concerns about wastewater irrigation is the risk of introducing pathogens to the soil and crops [19], its influence on soil properties, crop yield, and GHG emissions should also be considered [20].

Localized irrigation techniques in which water is added directly to the root zone, such as drip irrigation, have become increasingly popular for saving water by reducing deep percolation, surface runoff, and evaporation [21]. Fertigation, a technique in which nutrients and water are applied simultaneously to the crop with great precision in time and space, provides a technical solution to both deficiencies [22]. A study conducted in a semiarid climate with horticultural crops showed that combining drip irrigation with organic fertilizers could be a good management strategy for an additional reason: it mitigates N_2_O emissions compared to conventional irrigation [23].

Replacing traditional irrigation and fertilization with practices that reuse water and nutrients could have numerous environmental benefits. However, most evaluations only consider improvements in soil properties or crop yield and often do not assess the impact on GHG emissions, which is poorly understood. Against this background, the main aim of the present study was to assess the soil properties, crop yield, and GHG emissions of a fertigated maize crop with a chemical N application (urea) and an organic amendment (treated dairy effluent), both applied with N isodose under a subsurface drip irrigation system. The starting hypotheses were that: (1) fertigation treatments with N isodose have equal crop yields; (2) GHG emissions during the fertigation period are larger from the maize fertigated with dairy effluent than from urea; and (3) treated dairy effluent improves soil properties more than urea treatment.

## 2. Materials and Methods

### 2.1. Experimental Soil and Crop Management

The present study was conducted at the INTA Agricultural Experimental Station Hilario Ascasubi (EEA INTA H. Ascasubi; 39°23′ S, 62°37′ W, and 13 m above sea level), in the southern part of the Pampean Region, Argentina (Appendix A). The soil at the field site is an Entic Hapludoll, sandy loam (USDA Soil Taxonomy), within the “La Merced” soil series [24], typical of the area.

The experiment was performed from November 2020 to April 2021 (spring-summer seasons, SH) in a no-tillage farm of 0.35 ha that began to produce five years before this trial. The preceding summer crops were maize (*Zea mays* L.) (2019/2020) and onion (*Allium cepa* L.) (2018/2019). Between crops, during fallow periods, the soil was kept undisturbed and without irrigation. Maize (*Zea mays* L.) DK72-10VT3P was mechanized direct-seeded (no-tillage) on 4 November 2020 using a pneumatic seed metering (Figure 1a). Seeds had a germination power of 98.7% (data provided by the Seed Laboratory EEA INTA H. Ascasubi). Plant density was 90,000 pl ha^−1^ and spacing between rows was 0.70 m. Diammonium phosphate (DAP: 18-46-0) was banded with the seed at planting at 84 kg ha^−1^ laterally to the seeding row (37 kg P ha^−1^).

### 2.2. Weather Conditions and Subsurface Drip Irrigation

The experimental site is located in a cold semiarid steppe according to the climate classification (Köppen). The mean annual temperature is 15 °C, with an annual average precipitation of 492 mm and approximate annual Penman–Monteith evapotranspiration (ET) of 1080 mm [25]. The greatest rainfall events occur during the spring and summer months, and the area has a water deficiency of 300 to 500 mm, making it necessary to irrigate crops permanently. The mean temperature (20 °C) during the experiment was similar to the historical mean temperature (1966–2020). Rainfall showed a marked decrease compared to the historical average in the evaluated period (Table 1). The accumulated rainfall was higher than the historical average value only during December (Appendix A). The values of evapotranspiration (ET) were considerably lower during a large part of the maize cycle, with emphasis on the months of the maximum water demand of the crop.

The plot was irrigated through a subsurface drip irrigation (SDI) system. Lateral lines were beneath the plant rows at 0.25 m deep with emitters 0.5 m apart along laterals (Figure 1b,c). The drip system emitter flow rate was 1.6 L h^−1^ (DripnetPC, Netafim®, Hatzerim, Israel), delivering an irrigation rate of 4.57 mm ha^−1^. Water quality showed (Table 2) a low SAR ratio and a moderate risk of salinization [26]. Irrigations were scheduled using the daily corn water use obtained by the reference ET calculated using the Penman–Monteith equation with the available data from the meteorological station site and the local basal crop coefficients based on similar procedures outlined by Sanchez [27].

### 2.3. Experimental Design and Treatments

The experiment was arranged according to a complete block design with three treatments and four replicates. Treatments included: liquid fertigation with dissolved granulated urea (46-0-0; U), liquid fertigation with treated farm-dairy effluent (DE), and a control (C) irrigated without N fertilizer. The surfaces of the experimental units for the U and DE were 56 m^2^ (four rows of maize 20 m long for each fertigation treatment), separated by two rows of maize without fertigation (used as control). The GHG and soil samples were extracted from the center of each experimental unit.

Fertilizer application began almost a month after sowing and was divided into six fertigation events (once per week) to supply a total equivalent dose of 200 kg N ha^−1^. There was a first application on 1 December 2020 (day 0) and then additional applications on days 8, 14, 21, 28, and 38 after the first fertigation. This period lasted from maize growth stage V3 (third leaf) to R1 (silk emergence) [28]. Liquid DE and U treatments were applied through the SDI system. Fertigation with U was carried out with a hydraulic piston injector pump (Amiad^®^) into the SDI system, whereas the DE was filtered (ring filter, 120 mesh) and injected with a centrifugal pump directly into drip line irrigation (91,000 L ha^−1^ per application). Each application was computed as part of the irrigation to level the irrigation water dose between treatments.

Simple soil samples (0–10, 10–20, and 20–60 cm) for characterization were taken at two different times, before fertilizer applications (V2) and at the end of the ontogenic cycle (harvest). The soil was air-dried and sieved (<2 mm) for chemical analyses. Electric conductivity (EC) and pH were measured in soil-water extract (1:2.5) by potentiometry [29] and salinity was estimated in the saturated extract following the process by [30]. Organic matter (OM) was determined by weight loss in a muffle at 550 °C for 2 h [31], while extractable P (eP) was measured by colorimetry following Bray I extraction [29]. Additionally, soil samples were taken (0–10 cm) for ammonium (NH_4_-N) and nitrate (NO_3_-N) determination by semi-micro-Kjeldahl [32] on days 0, 14, 38, and 62 after the first fertigation event.

At the end of the experiment (15 April 2021), crop yield was estimated by collecting maize at the physiological maturity stage from a central area of 3.5 m^2^ on each parcel (n = 12) and then it was extrapolated for each treatment to kg ha^−1^ on a dry basis (forced air oven at 60 °C until reaching constant weight).

### 2.4. Dairy farm Description

The treated DE was collected from a commercial dairy farm located near Mayor Buratovich (39°17′ S, 62°32′ W), Buenos Aires Province, Argentina. A detailed description of the system was previously presented in Iocoli et al. [33]. Effluents produced during the cleaning of the milking area were collected in a series of 3 lagoons (the first anaerobic and the next two aerobic). The DE was collected from the last aerobic lagoon (as final product).

Chemical determinations for the DE characterization of pH, EC, NO_3_-N, NH_4_-N, and Total P (Table 3) were performed similarly to the procedures described above for the soil, with the exemption of the pH and EC which were run without dilution. Total Kjeldahl N (TKN) was determined by semi-micro-Kjeldahl [32]. The total C of the DE was determined by dry combustion at 1500 °C with a LECO C Analyzer (LECO Corporation, St. Joseph, MI, USA). To prepare the liquid DE, a 6 cm^3^ aliquot of each sample was added to the LECO inert absorbent material and oven-dried at 40 °C, following manufacturer recommendations. The total solids (TS) were determined by drying samples at 105 °C to constant weight and the ash content was determined by burning samples at 550 °C for 2 h, whereas the volatile solids (VS) were calculated by subtracting the ash percentages from the TS percentages.

### 2.5. GHG Measurements

Greenhouse gas fluxes were measured using the static chamber technique [34,35] from 1 December 2020 to 1 February 2021 (62 days). Twelve cylindrical PVC bases (30 cm internal diameter, 15 cm height, and area 0.07 m^2^) were used as the bottom of the static chambers. Chamber bases were inserted 10 cm into the ground in the center of each experimental unit (between maize rows) two days before treatment application to prevent soil disturbance from influencing gas emissions and left in place throughout the monitoring period. A lid of the same material and diameter as the bases (10 cm height) was used during sampling to enclose the internal volume of the chamber. The lid was equipped with a septum for sample extraction and a digital thermometer to record the internal air temperature during gas collection to correct flux calculation.

Periodic gas sampling began on the first day of fertigation (day 0) and ran until day 62 after application, when all treatments equaled background fluxes. Gas sampling was carried out three hours after each of the six fertigation events. In addition, in the first and last fertigation, more sampling campaigns were performed (1, 3, and 5 days after the fertigation event). More sampling frequency would be suitable for a better GHG emission estimation in the intermediate fertigations, but we had logistics limitations caused by the COVID-19 pandemic. Thus, sampling frequency varied throughout the experiment with a total of 15 sampling campaigns on days 0, 1, 3, 5, 8, 14, 21, 28, 38, 39, 41, 44, 48, 53, and 62.

The lids were fitted to the chamber bases on each sampling occasion and sealed with an airtight rubber belt. Samples were collected by a 25 mL polypropylene syringe after chamber closure (0 min) and after 15, 30, and 45 min to ensure an adequate linear response in headspace gas concentrations. Once sampling was completed, the chamber tops were removed and chamber bases left uncovered, exposing the soil to natural incident rainfall and solar radiation. Samples were collected and immediately transferred to pre-evacuated 12 mL (Labco Exetainer) vials.

Samples were analyzed by a gas chromatograph (GC Agilent 7890A) equipped with an electron capture detector (ECD) for N_2_O analyses, with a temperature of 280 °C and a carrier gas flow (Ar/CH_4_, 5%) of 2 mL min^−1^. The GC was also equipped with a flame ionization detector (FID) for CH_4_ and CO_2_ with a 1–8 Porapak Q column (80/100 mesh), and an oven, injector, and detector (temperatures of 60, 375, and 300 °C, respectively). The carrier gas (N_2_) was maintained at a constant pressure of 27 psi and the flow of flammable gases (H_2_ and air) was 40 and 450 mL min^−1^, respectively. The GC concentration was determined from 6-point calibration curves (R^2^ = 0.99) ranging from 0.17 to 1.38 ppm of N_2_O, 0.2 to 3.1 ppm of CH_4_, and 95 to 1579 ppm of CO_2_.

Gas fluxes were calculated as the rate of change in concentration over time using linear regression, described in detail in Lombardi et al. [36]. Fluxes were corrected with the chamber headspace, the internal air temperature at sampling and the atmospheric pressure at the site (from the nearest meteorological station). All flux data were checked for linearity by examining the R^2^, chambers were excluded from the dataset at R^2^ < 0.8. In addition, we assumed the absence of flux in chambers (i.e., flux = 0) when the rate of change in gas concentration was below the analytical precision limit of the GC device. This limit was determined following the procedure of Parkin et al. [37] by analyzing several ambient air samples and then calculating their coefficients of variation (3.8% for N_2_O, 2.1% for CH_4_, and 3.1% for CO_2_).

Cumulative fluxes (mg m^−2^) were calculated for N_2_O and CH_4_ by integrating the linear interpolation of the daily fluxes over time (OriginLab 2018 Software), for the first and last of the six fertigation periods when sampling was more frequent. Then, it was assumed that the four intermediate irrigations had emissions equal to the average between the first and last 5-day irrigation periods. Thus, the total cumulative fluxes for the 62 days were estimated as the sum of the first and last individual emissions plus four times the average value corresponding to the four intermediate fertigation events. Then, cumulative N_2_O and CH_4_ emissions were converted to carbon dioxide equivalents (CO_2_eq) using the global warming potential (GWP) values of 273 and 27.2, respectively [38]. The measured CO_2_ fluxes were not suitable for calculating net GHG emissions since they include the soil heterogeneity respiration and root respiration.

Finally, the emission factors (EFs) for N_2_O, expressed as percentages of N emitted per unit of N added, were calculated by subtracting the cumulative N_2_O emissions in the control from the cumulative N_2_O emissions directly lost from the fertilizer used and then dividing by the amount of N applied by fertilizer. The EFs for CH_4_ (g CH_4_ kg^−1^ VS) were calculated only for the dairy effluent application as the direct CH_4_ emissions from the effluent minus the cumulative CH_4_ emissions from the control divided per unit of volatile solids applied [39].

### 2.6. Statistical Analysis

Data analysis was performed using Infostat^®^ (v. 2020), a statistical software linked to the R programming environment [40]. Daily GHG fluxes were analyzed using a generalized linear mixed model (*lmer* package) with the irrigation treatment and sampling occasions as fixed factors, while each field plot was considered as a random factor. Several models were assessed, and the most appropriate fit was selected according to the lowest Akaike information criterion. The other variables were analyzed with ANOVA and differences were determined using the LSD Fisher test at the *p* < 0.05 level. Correlations between soil parameters and GHG fluxes were determined using Spearman’s coefficient.

## 3. Results

### 3.1. Soil Parameters and Maize Yield

The results showed an initial edaphic condition with a slightly acidic upper layer pH of 6.6 with an increasing reaction in subsurface layers (Appendix A). The mean estimated EC of the saturated extract was higher in the 0–10 cm layer and varied between 1.50 and 1.85 dS m^−1^, which is considered relatively low and barely problematic for maize [41]. The high content of eP was measured in the 0–10 cm layer (31–49 mg kg^−1^) as a consequence of fertilization in previous production cycles. The OM content in the 0–10 cm layer was 2.65–2.99%.

No differences were detected in soil NH_4_^+^ and NO_3_^−^ levels between treatments (Figure 2). However, DE presented the highest NO_3_^−^ values at the end of the 62-day trial.

After the crop cycle, some significant differences in soil properties were detected. The C treatment was the only one with a pH increment trend, with a nonsignificant difference of +0.41 units in the 0–10 cm layer (*p* = 0.13). The general trend for DE and U pointed towards lower soil reactions and the only significant pH descent was found for the U treatment in the 10–20 cm layer at the end of the experience (*p* < 0.05). For the EC, there were significant increments only comparing the initial and the final conditions, but there were no differences between treatments. The EC showed an increment for DE in the 0–10 cm layer (from 1.50 to 2.12 dS m^−1^, *p* < 0.05) and 20–60 cm layer (from 1.45 to 1.81 dS m^−1^, *p* < 0.05), while the middle layer (10–20 cm) only showed a nonsignificant increasing trend for DE (*p* = 0.28). For the U treatment, an increase in EC was also observed in the 10–20 cm layer from 1.27 to 1.57 dS m^−1^ (*p* < 0.05). There were no significant changes for eP, even though there was a trend of lower values at harvest in all treatments, showing active consumption of this element. For OM, the comparison between the initial and final conditions exhibited a significant increase in the 20–60 cm layer for DE (*p* < 0.01) and U (*p* < 0.05). The changes in OM concentration (end-start condition) showed that in the 10–20 cm layer, DE and U treatments slightly increased their OM values and therefore behaved significantly different from C, which diminished their OM values (*p* < 0.05). A similar trend was also observed in the 0–10 cm layer but was not significant (*p* = 0.29). All these changes pointed to a slight increase in the OM in the DE and U treatments, especially in the 10–60 cm depth.

No significant differences in maize yield were found across the treatments; nevertheless, there was a clear tendency to higher values in U (16,509 ± 1181 kg ha^−1^) and DE (16,270 ± 1774 kg ha^−1^) compared to in treatment C (14,957 ± 1335 kg ha^−1^).

### 3.2. GHG Fluxes

The three GHG fluxes measured in the soil surface of the maize crop across the entire 62-day monitoring period were influenced by the treatment (*p* < 0.0001), the time after treatment application (*p* < 0.0001), and their interaction (*p* < 0.0001). This significant interaction indicates that the treatment had a different effect on GHG emissions depending on the sampling occasion. Due to the performed sampling frequency, a continued flux pattern can be seen in Figure 3 on the first and last fertigation periods, whilst in the four intermediate sampling occasions, it shows the instant fluxes after 3 h on each fertigation event. No general trends were observed while assessing the correlations between GHG fluxes and soil properties (such as soil moisture or NH_4_^+^ and NO_3_^−^ content).

The DE treatment had the highest N_2_O fluxes (68.5 µg N_2_O-N m^−2^ h^−1^), followed by the U (23.5 µg N_2_O-N m^−2^ h^−1^) and C (16.8 µg N_2_O-N m^−2^ h^−1^) treatments (Figure 3b). A delayed N_2_O peak was observed from the DE treatment in both of the frequently measured periods (first and last fertigation), whilst the U treatment only showed an N_2_O peak on the last fertigation occasion. From day 44 onwards, one week after the last fertigation, there were no statistically significant differences between treatments.

The highest CH_4_ fluxes were measured from the DE treatment (3.39 mg CH_4_-C m^−2^ h^−1^), which immediately increased after three hours on each of the six fertigation events and then decreased (Figure 3c). For the rest of the following sampling occasions, the DE fertigation showed negative CH_4_ fluxes similar to those found from C and U treatments. There were no significant differences between U and C treatments across the 62-day monitoring period. The mean CH_4_ flux from both U and C was −0.03 mg CH_4_-C m^−2^ h^−1^, acting as a sink for CH_4_. From day 39 onwards, when fertigation stopped, the CH_4_ fluxes were similar between the three irrigation treatments.

The CO_2_ fluxes from the U and C treatments showed no significant differences across the entire monitoring period. The DE treatment remained similar to the control, except for the irrigation events when significant CO_2_ emissions were stimulated immediately after 3 h of being applied (Figure 3d). Then, CO_2_ fluxes from DE decreased to reach C values, similar to the CH_4_ emission pattern from DE. The maximum CO_2_ fluxes in the DE were 123 mg CO_2_-C m^−2^ h^−1^, whilst the C and U treatments peaked at 81.2 and 85.8 mg CO_2_-C m^−2^ h^−1^, respectively. From day 39 onwards, there were no significant differences among irrigation treatments on the CO_2_ fluxes.

Estimated cumulative fluxes over the 62-day period varied depending on the treatments applied (Table 4). The DE showed significantly higher N_2_O cumulative fluxes, followed by the U treatment, both significantly higher than C. The DE treatment also resulted in a significant positive value (emission of CH_4_), whilst the CH_4_ cumulative fluxes from the C and U treatments were negative (sink for CH_4_). Finally, there were no significant differences in the net CO_2_eq emissions from the U and C treatments; meanwhile, DE resulted in the significant highest CO_2_eq values.

The EF for N_2_O from the DE fertigation was 0.11 ± 0.03%, significantly higher than the 0.03 ± 0.01% from the U fertigation (SEM = 0.01, *p* = 0.003, and n = 4). The CH_4_ emission factor (EF) for the DE treatment was 8.1 ± 3.1 g CH_4_ kg^−1^ VS.

## 4. Discussion

### 4.1. Effect of Fertigation Treatments on Soil Properties and Maize Yield

The liquid DE was rich in N, C, and P, as was observed in several studies despite the DE originating from different farm systems, and had different physical, chemical, and biological characteristics [42,43,44]. A previous spectroscopic characterization performed in the same DE used in the present study suggested that DE was composed of soluble and amino sugars, amino and short-chain organic acids, and proteins that may be rapidly degraded after application in the soil [33]. Thus, in general, it has been observed that the application of liquid organic amendments to the land improves soil fertility and increases the efficiency of the use of resources in farming systems [42]. However, in this study, only minor changes in soil properties were observed since only one fertigation period was assessed, and more fertigation cycles would be necessary for detecting long-term improvements in soil fertility.

In this study, changes in EC and OM were observed, while for pH and eP no definitive trends were found. Soil EC showed an increase in the 0–10 and 20–60 cm layers for DE and at 10–20 cm for U. This could indicate higher fertility or salinity depending upon the elements. The SDI system could develop salt accumulation around the emitters and especially near the soil surface due to direct evaporation [45]. For DE, there was also an additional salt input due to the high salinity content of the DE used (4.73 mS cm^−1^, Table 3) explaining the observed difference. The EC increase is more evident when low amounts of precipitation occur that prevent salt leaching. This could be the reason for the increased EC, with some salt accumulation that could not be leached due to the low rainfall during the 2020/2021 season (Table 1). This salt excess is expected to decrease to the initial condition with normal precipitation, as observed by Varela et al. [46].

A slight but significant increase in OM was observed in the DE and U treatments for the 20–60 cm soil depth. The results of this indicator were difficult to find in the literature because the main objectives of similar experiences were mainly focused on maize yield. However, these results were similar to those obtained by Mitchell [47], who found higher OM levels after two cycles of a maize crop with SDI. Those results were attributed to sampling some root biomass growing around the wetting zone generated by the emitters. A higher root development could be expected with higher grain yields explaining the results obtained here. However, other authors indicate that roots tend to grow concentrated near the water source, diminishing the soil volume explored [48]. Nevertheless, after a single maize crop, it is still too early to indicate if our results point to a random fresh OM (roots) sampling or a stable increase in OM in the soil. More production cycles would be needed to detect middle- to long-term stable OM increments due to the fertigation.

Similar maize yields were obtained from both the U and DE fertigation treatments, as was expected since both were applied an isodose of N. In fact, the maize yields of the three treatments were similar and without statistically significant differences. The average of all treatments (15.5 Mg ha^−1^) exceeded the local average yield (8 Mg ha^−1^) for the maize crop that is commonly irrigated under gravitational systems [45]. The greatest advantage of the SDI system could be explained not only by the high capability of satisfying the water supply during the critical period VT-R2 [41] but also because of the fertigation process that delivers the fertilizer accurately and reduces the risks of nutrient loss. Furthermore, the benefit of fertigation in crops such as maize is to allow fertilization even in the advanced phenological stage (R1) [47], in which the maize is still uptaking N and where the traffic with conventional fertilizing machinery would not be possible because of the mechanical damage over plants.

The high performance of the C treatment was likely due to the characteristics of the site. It was never laser land leveled (not cut or filled) and it began its productive use five years before this trial, always managed under direct sowing (no-tillage) and systematized with SDI. Consequently, the soil maintained higher values of OM and nutrients than the surrounding productive lots of the region which were under gravitational irrigation (1–2% [25]). In addition, the dose used (200 kg N ha^−1^) was lower than that necessary to express the maximum regional yield of the hybrid (300 kg N ha^−1^; the production of 18–20 Mg of maize [49]), but it was defined according to the operational possibilities for the application of the DE. If a higher dose of N had been applied, the differences between the maize yields of the DE and U plots compared to C would have been greater.

Although similar maize yields were obtained from both U and DE, the primary N source used in Argentina for agricultural production is urea despite its high costs. While the national production of fertilizers covers barely 30% of the annual crop demand, the rest is imported. The urea represents 40% of the total fertilizers used in Argentina [50] and the dependency on the international market makes the urea into an input with a fluctuating value that conditions the cost of crop production. Thus, the adoption of SDI systems combined with the reuse of effluents from intensive dairy farms can constitute a promising strategy to increase maize performance at a lower cost [48,49] while reducing the environmental impact of waste accumulation and industrial fertilizers production.

### 4.2. Effect of Fertigation Treatments on GHG Emissions

The application of DE in the maize crop had the highest net GHG emissions during the fertigation period, as expected. This effect was also reported by Vico et al. [35] comparing organic-based stabilized materials with inorganic fertilizers for intensive spinach production. The higher emissions from the DE could be attributed to the microbial stimulation promoted by the greater amount of available nutrients of the DE during the fertigation. In general, the land application of liquid manure (like pig slurries or dairy effluents) enhances soil CH_4_ and CO_2_ fluxes immediately after application due to the introduction of substrate for soil-borne microbes such as mineral N (NH_4_^+^ and NO_3_^−^) and easily mineralizable N and C [51,52]. This increases microbial respiration (CO_2_) which, combined with high moisture and low gas diffusivity, creates anaerobic conditions near subsurface irrigation lines, raising soil CH_4_ fluxes [13,44]. These areas may favor nitrification, nitrifier denitrification, and subsequently denitrification, increasing soil N_2_O fluxes [12,53]. Through these same processes, N_2_O was emitted after U application but at a lower rate than the DE despite applying the same amount of N. In addition, the lack of correlation between NO_3_^−^ and NH_4_^+^ levels and N_2_O emissions may indicate that the DE enhanced N_2_O emissions more than U due to the increased content of readily available organic C compounds and/or from decreased soil aeration following increased respiration.

The N_2_O peak emissions from the DE and U were higher in the last fertigation event (Figure 3b). Likely the accumulation of N derived from the successive fertigations could originate the delayed N_2_O emissions. In addition, in the DE treatment, the slow degradation of the more complex N compounds could also delay the N_2_O emissions. In the present study, 34% of the total N in the DE was in the form of readily available mineral N (NH_4_^+^ and NO_3_^−^). The remaining 66% was in the organic form which was not immediately available [42]. Thus, the N fractions in DE are a mixture of fast- and slow-release compounds, which sustain both short- and long-term responses to the effluent application [44]. This long-term response was also supported by the higher NO_3_^−^ content in the DE soil at the end of the 62-day period compared to the beginning of the experiment (40 vs. 27 mg kg^−1^; respectively). In addition, at the end of the gas measurement period, the NO_3_^−^ content in DE soil was almost double the NO_3_^−^ in C and U soils (40 vs. 24 mg kg^−1^), whereas the NH_4_^+^ contents of DE and U were similar to that of C (53 mg kg^−1^).

Under the present experimental conditions, the daily irrigation (between 7 and 13 mm d^−1^) in the last two weeks did not significantly increase the N_2_O fluxes (neither CH_4_), probably indicating that it derived directly from the fertigation events. If upcoming environmental conditions are favorable, it could lead to late N_2_O emissions. However, these emissions are expected to be low and short in time since the N content in soil was low and no more fertilizer was added.

The maximum CH_4_ fluxes were measured immediately after each DE application, above 1 mg CH_4_-C m^−2^ h^−1^, while the U and C treatments remained the entire measurement period below −0.01 mg CH_4_-C m^−2^ h^−1^. This behavior was also reported by Viguria et al. [48] after applying cattle slurry to bare soil. According to these authors and another previous work [54], these CH_4_ peaks after manure incorporation into soils are generally short-lived due to the diffusion of oxygen into the manure products and the subsequent inhibition of CH_4_ formation. Moreover, the main factor determining the extent of CH_4_ produced in the effluents is the amount of degradable OM [44], which might be why CH_4_ emissions were observed only from the DE. Other authors related the immediate high CH_4_ emissions after application to the release of entrained CH_4_ produced during DE storage [55]. According to Le Mer and Roger [56], the methanogenic fermentation of organic amendments occurs under strictly anaerobic and low-redox-potential (Eh < −200 mV) conditions, where NO_3_^−^ contents are low, similar to conditions near the subsurface irrigation lines during the fertigation events of this experiment. This could explain why the maximum CH_4_ fluxes decreased from 3.8 to 1.1 mg CH_4_-C m^−2^ h^−1^ throughout the fertigation events.

Therefore, the application of DE emitted more CO_2_eq than that of U during the 62-day fertigation period. However, to compare treatments and production cycles, some considerations for GHG accounting must also be taken into account. On the one hand, the reuse of DE can mitigate the environmental impact of intensive dairy farms by reducing the liquid effluent’s retention time, which greatly affects the amount of CH_4_ produced [7]. Considering the VS content of the DE used and the application rate, 256 kg VS were removed from the dairy farm to fertigate a hectare of maize. Therefore, theoretically, a minimum of 850 kg CO_2_eq could be mitigated in one year by removing that amount of effluent from the dairy farm, based on the CH_4_ EF proposed by IPCC [39] for uncovered lagoons on high-productivity dairy systems (122.2 g CH_4_ kg VS^−1^). This was five times greater than the 150 kg CO_2_eq emitted per hectare by the use of DE during the fertigation period. Thus, regardless of using U or not, simply applying the effluent into the soil would reduce the potential GHG emissions of the dairy system. In addition, no GHG would be emitted to obtain the DE with which the maize was fertigated since it was reused from the dairy farm, turning a waste for disposal into a resource.

On the other hand, to obtain the U, like other industrial fertilizers, an energetically and economically expensive process must be carried out. It consumes about 10–19 KWh kg^−1^ N produced by fixing atmospheric N_2_ through the Haber–Bosch process [57,58]. Some studies indicated that this process could emit between 3.8 and 5.8 kg CO_2_eq kg^−1^ N [59,60]. At a rate of 200 kg N ha^−1^_,_ as used in this experiment, the theoretical emission of CO_2_eq from U manufacturing would be between 760 and 1160 kg CO_2_eq for the U applied in a hectare. Again, these values are considerably higher than the CO_2_eq emissions per hectare from either the DE or U used during the fertigation period. Other studies also found a reduction in CO_2_eq emissions as a consequence of fertigating with organic amendments due to OM revalorization, which also helps conserve freshwater sources and reduces the energy consumed for industrial fertilizer production because of the total or partial avoidance of mineral N fertilizer requirement [60,61]. Therefore, these values should be taken into account when comparing fertilization treatments. In addition, financial benefits for farmers can also be expected from savings in mineral fertilizers [62].

This study revealed that EFs for N_2_O from a maize crop with SDI during fertigation averaged 0.07 ± 0.05% regardless of the fertilizer used. This value was substantially lower than the IPCC default value of 1% [63] and the updated IPCC default value of 0.5% for dry climates [39]. Considering the type of fertilizer, EFs for N_2_O were higher from the DE (0.11 ± 0.03%) than from the U (0.03 ± 0.01%). Similarly, a meta-analysis of N_2_O EFs from Mediterranean climate cropping systems found that organic liquid fertilizers, which mainly were pig or cattle slurries, or the liquid fraction of their digestates, had EF values (0.85%) almost double the EFs from synthetic fertilizers (0.45%) [64]. Conversely, another meta-analysis performed with data from New Zealand found a lower mean EF for N_2_O from the farm-dairy effluent application (0.3%) compared to urea application (0.6%) [64]. In addition, Cayuela et al. [65] found that there was high variability in EFs between the types of irrigation management and that the EFs for N_2_O from drip-irrigated systems (0.51%, including both surface and subsurface and all types of fertilizers) were much lower than those from systems with sprinklers (0.91%). Other studies found similar results comparing SDI systems with other conventional irrigation systems [23,66]. It seems that the SDI system used in the present study can lead to low GHG emissions regardless of the fertilizer used, although further studies are needed since there is little information about it. These findings should be evaluated with other effects on the GHG balance and further socioenvironmental consequences.

## 5. Conclusions

The results obtained have shown that fertigation using dairy effluent may have the potential to increase soil fertility, also being as efficient as urea concerning maize yield. The application of dairy effluent increased net GHG emissions more than the urea during the fertigation period, probably due to the higher availability of nutrients for microorganisms in the dairy effluent. However, the reuse of the dairy effluent can constitute a promising alternative to chemically synthesized fertilizers such as urea, contributing to reducing the environmental impact of intensive dairy production systems while simultaneously avoiding the theoretical CO_2_eq emissions from energetically expensive urea manufacturing. On the other hand, the considerably lower N_2_O emission factor from the fertigation with subsurface drip irrigation systems (0.07%), compared to the default IPCC values (1 and 0.5% of guidelines 2006 and 2019, respectively), confirmed the need for local-management-specific emission factors, representative of site semiarid conditions. Additional research should be performed to better understand the efficiency of the different irrigation systems, while combining fertilizers from different sources (e.g., mix of dairy effluent with urea), to find optimal nutrient use efficiency and lower environmental impacts.

## Figures and Tables

**Figure 1 animals-12-02025-f001:**
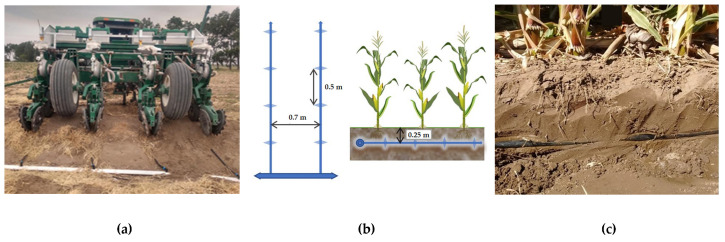
(**a**) Mechanized seeding, rear view on one of the headers of the field; (**b**) schematic view (top and transverse) of the subsurface drip irrigation system (SDI); (**c**) soil profile with the irrigation line at 0.25 m depth.

**Figure 2 animals-12-02025-f002:**
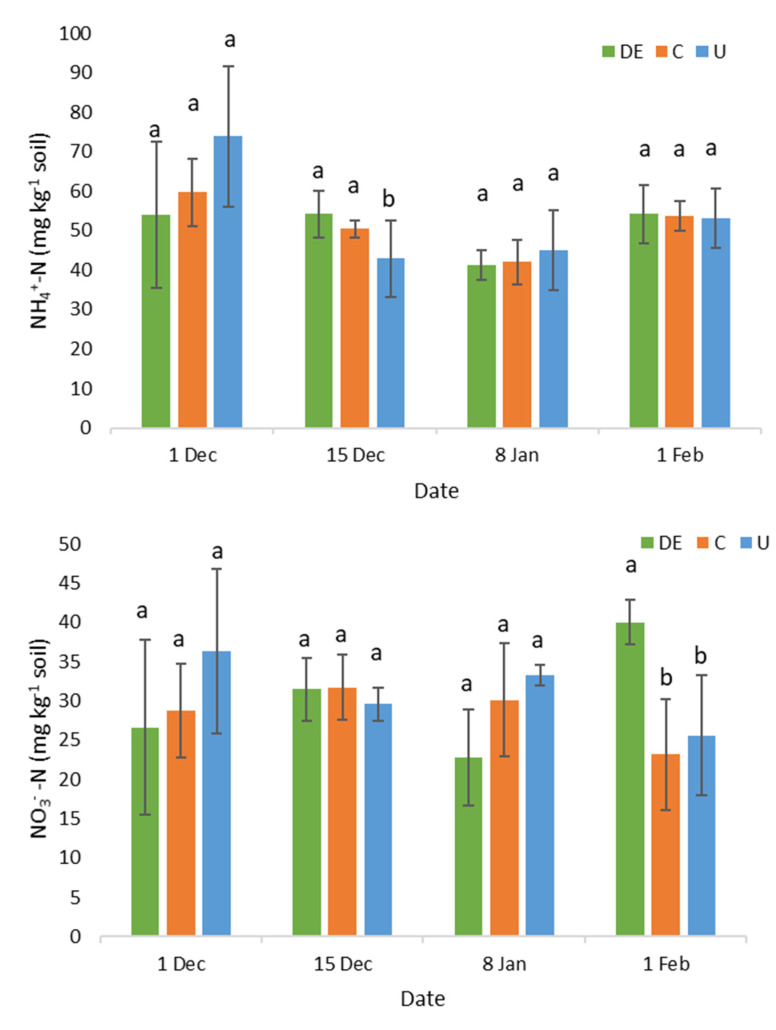
NH_4_^+^-N (**top**) and NO_3_^−^-N (**bottom**) levels after fertigation. Different letters indicate significant statistical differences (LSD *p* < 0.05) within each date. DE: dairy effluent (green); C: control (orange); U: urea (blue).

**Figure 3 animals-12-02025-f003:**
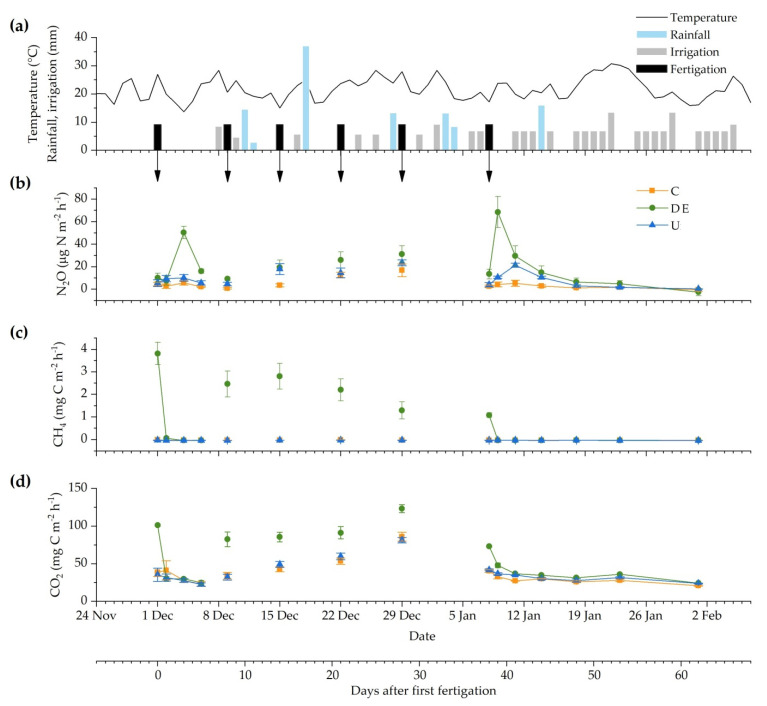
(**a**) Mean air temperature (°C) and rainfall, irrigation, and fertigation (mm). Daily fluxes of (**b**) N_2_O, (**c**) CH_4_, and (**d**) CO_2_ from the control (C), dairy effluent (DE), and urea (U) application during the 62-day monitoring period. Arrows indicate fertigation events. The bottom axis shows the days after the first fertigation (day 0) and corresponding dates. Bars represent the standard error of the mean.

**Table 1 animals-12-02025-t001:** Historical rainfall and rainfall, evapotranspiration (ET), irrigation, and water balance (in mm) throughout the maize irrigation cycle.

	Historical Rainfall	Rainfall	ET	Irrigation	Balance
Months			(mm)		
November	44.0	39.7	47.4	0.0	−7.7
December	52.0	68.2	149.2	81.6	0.6
January	49.4	37.6	190.4	142.0	−10.8
February	55.9	2.0	134.2	136.0	3.8
March	62.4	18.3	29.5	20.4	9.2
TOTAL	263.3	215.3	562.7	380.0	32.6

**Table 2 animals-12-02025-t002:** Physicochemical characteristics of the irrigation water used in the experiment.

pH	EC	Na^+^	HCO_3_^−^	Cl^−^	SAR
	(dS m^−1^)	(meq L^−1^)	(meq L^−1^)	(meq L^−1^)	
8.21	1.89	7.61	1.87	6.58	3.87

EC: electrical conductivity; SAR: sodium adsorption ratio.

**Table 3 animals-12-02025-t003:** Physicochemical characteristics of the treated dairy effluent (DE).

	pH	EC	C	TKN	NO_3_^−^-N	NH_4_^+^-N	eP	TP	TS	VS
		(dS m^−1^)	(mg L^−1^)							
DE	7.8	4.73	663	672	10.10	215	1.52	111	3070	859

EC: electrical conductivity; TKN: total Kjeldahl N; NO_3_^−^-N: nitrate; NH_4_^+^-N: ammonium; eP: extractable phosphorus; TP: total phosphorus; TS: total solids, VS: volatile solids.

**Table 4 animals-12-02025-t004:** Estimated cumulative N_2_O and CH_4_ fluxes during the 62-day measuring period.

	N_2_O	CH_4_	CO_2_eq ^1^
Treatment	(mg m^−2^)	(mg m^−2^)	(kg ha^−1^)
Control	5.3 a	−42 a	2.9 a
Dairy Effluent	38.5 c	165 b	149.9 b
Urea	14.5 b	−53 a	25.1 a
SE^2^	2.7	22	12.3
*p*-value	<0.0001	0.0001	<0.0001

^1^ CO_2_ equivalent from added N_2_O and CH_4_ emissions with the corresponding GWP [35]. ^2^ Standard error. Treatments with different letters within the same column represent significant differences at *p* < 0.05, tested separately for each gas.

## Data Availability

Not applicable.

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
