# Peer review of "Is Dairy Effluent an Alternative for Maize Crop Fertigation in Semiarid Regions? An Approach to Agronomic and Environmental Effects"

_animals, 2022, doi:10.3390/ani12162025_

Round 1
Reviewer 1 Report
Dear authors: I want to mention that the topic you are presented is very interesting and important to have an alternative for soil treatment and reuse of agricultural waste, it is only recommended that you make the information obtained more attractive to the reader because the importance of your results is very important to highlight so that more people can understand the manuscript.
Reviewer 2 Report
In the manuscript, the authors assess greenhouse gas (GHG) emissions, soil properties and crop yield of a maize crop fertigated with either treated dairy effluent or dissolved granulated urea applied through an SDI system. The authors present interesting results and the manuscript can be accepted after minor revisions.
Specific comments:
1, The introduction section can be refined.
2. Fig S2 can be moved to the main text.
3, More literatures from the current journal can be included in the reference list.
Reviewer 3 Report
General comments:
The submitted article represents field measurements of greenhouse gas (GHG) emissions, soil properties, and crop yield (maize) fertigated with urea (as conventional nitrogen fertilizer) and treated dairy effluent (as an organic amendment) through a subsurface drip irrigation system. The manuscript is, in general, well written, and the findings are interesting. However, a few issues need to be addressed prior to its further processing.
Specific comments:
1. The authors used the word 'dairy' and 'cattle' interchangeably in some places (L 54, 67, 472, 520). It would be better to use the word 'dairy' in all cases.
2. The novelty statement should be more robust.
3. Section 2.1: It would be better to include the climatic characteristics of the study site.
4. L293: Mention the p-value. Check for such issues throughout the result section.
5. Figure 3: Check the figure number. It should be Figure 2. Also, replace the figure with a higher resolution one.
6. Section 4.1: It is well established that dairy manure is a good source of nutrients (nitrogen, N and phosphorus, P) for agricultural crops, but manure N:P ratios do not often match crops' needs of N:P ratios. In most cases, when manure is directly applied to crops that can't fix N from the air and if the manure application rate is based on crops' N needs, P is over-applied (also done in this study). Therefore, the potential P runoff from the manure over-supply to crop fields has further several environmental implications. The authors need to consider this aspect in their study, especially when highlighting the benefits of land application of treated dairy effluent.
7. L374-375: Why did the authors consider only one fertigation period to assess changes in soil properties? Consider adding an explanation.
